# Temperature explains broad patterns of Ross River virus transmission

Marta Strecker Shocket[1]*, Sadie J Ryan[2,3,4], Erin A Mordecai[1]

[1]Department of Biology, Stanford University, Stanford, United States; [2]Department of Geography, University of Florida, Gainesville, United States; [3]Emerging Pathogens Institute, University of Florida, Gainesville, United States; [4]School of Life Sciences, College of Agriculture, Engineering, and Science, University of KwaZulu Natal, KwaZulu Natal, South Africa

**Abstract** Thermal biology predicts that vector-borne disease transmission peaks at intermediate temperatures and declines at high and low temperatures. However, thermal optima and limits remain unknown for most vector-borne pathogens. We built a mechanistic model for the thermal response of Ross River virus, an important mosquito-borne pathogen in Australia, Pacific Islands, and potentially at risk of emerging worldwide. Transmission peaks at moderate temperatures (26.4°C) and declines to zero at thermal limits (17.0 and 31.5°C). The model accurately predicts that transmission is year-round endemic in the tropics but seasonal in temperate areas, resulting in the nationwide seasonal peak in human cases. Climate warming will likely increase transmission in temperate areas (where most Australians live) but decrease transmission in tropical areas where mean temperatures are already near the thermal optimum. These results illustrate the importance of nonlinear models for inferring the role of temperature in disease dynamics and predicting responses to climate change.
DOI: https://doi.org/10.7554/eLife.37762.001

*For correspondence:
mshocket@stanford.edu

**Competing interests:** The authors declare that no competing interests exist.

## Introduction

Temperature impacts the transmission of mosquito-borne diseases via effects on the physiology of mosquitoes and pathogens. Transmission requires that mosquitoes be abundant, bite a host and ingest an infectious bloodmeal, survive long enough for pathogen development and within-host migration (the extrinsic incubation period), and bite additional hosts—all processes that depend on temperature (*Mordecai et al., 2013*, *Mordecai et al., 2017*). Although both mechanistic (*Mordecai et al., 2013*, *Mordecai et al., 2017*; *Liu-Helmersson et al., 2014*; *Wesolowski et al., 2015*; *Paull et al., 2017*) and statistical models (*Perkins et al., 2015*; *Siraj et al., 2015*; *Paull et al., 2017*; *Peña-García et al., 2017*) support the impact of temperature on mosquito-borne disease, important knowledge gaps remain. First, how the impact of temperature on transmission differs across diseases, via what mechanisms, and the types of data needed to characterize these differences all remain uncertain. Second, the impacts of temperature on transmission can appear idiosyncratic—varying in both magnitude and direction—across locations and studies (*Gatton et al., 2005*; *Jacups et al., 2008a*; *Stewart-Ibarra and Lowe, 2013*; *Peña-García et al., 2017*; *Koolhof et al., 2017*). Although inferring causality from field observations and statistical approaches alone remains challenging, nonlinear thermal biology may mechanistically explain this variation. As the climate changes, filling these gaps becomes increasingly important for predicting geographic, seasonal, and interannual variation in transmission of mosquito-borne pathogens. Here, we address these gaps by building a model for temperature-dependent transmission of Ross River virus (RRV), the most important mosquito-borne disease in Australia (1500–9500 human cases per year) (*Koolhof et al., 2017*), and potentially at risk of emerging worldwide (*Flies et al., 2018*).

**eLife digest** Mosquitoes cannot control their body temperature, so their survival and performance depend on the temperature where they live. As a result, outside temperatures can also affect the spread of diseases transmitted by mosquitoes. This has left scientists wondering how climate change may affect the spread of mosquito-borne diseases. Predicting the effects of climate change on such diseases is tricky, because many interacting factors, including temperatures and rainfall, affect mosquito populations. Also, rising temperatures do not always have a positive effect on mosquitoes – they may help mosquitoes initially, but it can get too warm even for these animals.

Climate change could affect the Ross River virus, the most common mosquito-borne disease in Australia. The virus infects 2,000 to 9,000 people each year and can cause long-term joint pain and disability. Currently, the virus spreads year-round in tropical, northern Australia and seasonally in temperate, southern Australia. Large outbreaks have occurred outside of Australia, and scientists are worried it could spread worldwide.

Now, Shocket et al. have built a model that predicts how the spread of Ross River virus changes with temperature. Shocket et al. used data from laboratory experiments that measured mosquito and virus performance across a broad range of temperatures. The experiments showed that ~26°C (80°F) is the optimal temperature for mosquitoes to spread the Ross River virus. Temperatures below 17°C (63°F) and above 32°C (89°F) hamper the spread of the virus. These temperature ranges match the current disease patterns in Australia where human cases peak in March. This is two months after the country's average temperature reaches the optimal level and about how long it takes mosquito populations to grow, infect people, and for symptoms to develop.

Because northern Australia is already near the optimal temperature for mosquitos to spread the Ross River virus, any climate warming should decrease transmission there. But warming temperatures could increase the disease's transmission in the southern part of the country, where most people live. The model Shocket et al. created may help the Australian government and mosquito control agencies better plan for the future.

DOI: https://doi.org/10.7554/eLife.37762.002

RRV in Australia is an ideal case study for examining the influence of temperature. Transmission occurs across a wide latitudinal gradient, where climate varies substantially both geographically and seasonally. Moreover, compared to vector-borne diseases in lower-income settings, RRV case diagnosis and reporting are more accurate and consistent, and variation in socioeconomic conditions (and therefore housing and vector control efforts) at regional and continental scales is relatively low. Previous work has shown that in some settings temperature predicts RRV cases (*Gatton et al., 2005*; *Bi et al., 2009*; *Werner et al., 2012*; *Koolhof et al., 2017*), while in others it does not (*Hu et al., 2004*; *Gatton et al., 2005*). Understanding RRV transmission ecology is critical because the virus is a candidate for emergence worldwide (*Flies et al., 2018*), and has caused explosive epidemics where it has emerged in the past (infecting over 500,000 people in a 1979–80 epidemic in Fiji) (*Klapsing et al., 2005*). RRV is a significant public health burden because infection causes joint pain that can become chronic and cause disability (*Harley et al., 2001*; *Koolhof et al., 2017*). A mechanistic model for temperature-dependent transmission could help explain these disparate results and predict potential expansion.

Mechanistic models synthesize how environmental factors like temperature influence host and parasite traits that drive transmission. Thermal responses of ectotherm traits are usually unimodal: they peak at intermediate temperatures and decline towards zero at lower and upper thermal limits, all of which vary across traits (*Dell et al., 2011*; *Mordecai et al., 2013*; *Mordecai et al., 2017*). Mechanistic models are particularly useful for synthesizing the effects of multiple, nonlinear thermal responses that shape transmission (*Rogers and Randolph, 2006*; *Mordecai et al., 2013*). One commonly used measure of disease spread is $R_0$, the basic reproductive number, defined as the number of secondary cases expected from a single case in a fully susceptible population. Relative $R_0$— $R_0$ scaled between 0 and 1—is a modified metric that captures the thermal response of transmission without making assumptions about other factors that affect the absolute value of $R_0$ (*Mordecai et al., 2013*, *Mordecai et al., 2017*). For mosquito-borne disease, $R_0$ is a nonlinear

function of mosquito density, biting rate, vector competence (infectiousness given pathogen exposure), and adult survival; pathogen extrinsic incubation period; and human recovery rate (*Dietz, 1993*). To understand how multiple traits that respond nonlinearly to temperature combine to affect transmission, we incorporate empirically-estimated trait thermal responses into a model of relative $R_0$. Synthesizing the full suite of nonlinear trait responses is critical because such models often make predictions that are drastically different, with transmission optima up to 7°C lower, than models that assume linear or monotonic thermal responses or omit temperature-dependent processes (*Mordecai et al., 2013*, *Mordecai et al., 2017*). Previous mechanistic models that incorporated multiple nonlinear trait thermal responses have predicted different optimal temperatures across pathogens and vector species: 25°C for *falciparum* malaria in *Anopheles* vectors (*Mordecai et al., 2013*) and West Nile virus in *Culex* vectors (*Paull et al., 2017*), and 29 and 26°C for dengue, chikungunya, and Zika viruses (in *Aedes aegypti* and *Ae. albopictus*, respectively) (*Liu-Helmersson et al., 2014*; *Wesolowski et al., 2015*; *Mordecai et al., 2017*).

Here, we build the first mechanistic model for temperature-dependent transmission of RRV and ask whether temperature explains seasonal and geographic patterns of disease. We use data from laboratory experiments with two important vector species (*Culex annulirostris* and *Aedes vigilax*) to parameterize the model with unimodal thermal responses. We then use sensitivity and uncertainty analyses to determine which traits drive the relationship between temperature and transmission potential and identify key data gaps. Finally, we illustrate how temperature currently shapes patterns of disease transmission across Australia. The model correctly predicts that RRV disease is year-round endemic in tropical, northern Australia with little seasonal variation due to temperature, and seasonally epidemic in temperate, southern Australia. These results provide a mechanistic explanation for idiosyncrasies in RRV temperature responses observed in previous studies (*Hu et al., 2004*; *Gatton et al., 2005*; *Bi et al., 2009*; *Werner et al., 2012*; *Koolhof et al., 2017*). A population-weighted version of the model (assuming a two-month lag between temperature and human cases based on mosquito and disease development times) also accurately predicts the seasonality of human cases nationally. Thus, from laboratory data on mosquito and parasite thermal responses alone, this simple model mechanistically explains broad geographic and seasonal patterns of disease.

## Natural history of RRV

The natural history of RRV is complex: transmission occurs across a range of climates (tropical, subtropical, and temperate) and habitats (urban and rural, coastal and inland) and via many vertebrate reservoir and vector species (*Claflin and Webb, 2015*). Marsupials are generally considered the critical reservoirs for maintaining the virus between human outbreaks, but recent work has argued that placental mammals and birds may be equally important in many locations (*Stephenson et al., 2018*). The virus has been isolated from over 40 mosquito species in nature, and 10 species transmit it in laboratory studies (*Harley et al., 2001*; *Russell, 2002*). However, four species are responsible for most transmission to humans (*Culex annulirostris*, *Aedes [Ochlerotatus] vigilax*, *Ae. [O.] notoscriptus*, and *Ae. [O.] camptorhynchus*), with two additional species implicated in outbreaks (*Ae. [Stegomyia] polynesiensis* and *Ae. [O.] normanensis*).

The vectors differ in climate and habitat niches, leading to geographic variation in associations with outbreaks. We assembled and mapped records of RRV outbreaks in humans attributed to different vector species (*Figure 1*, *Figure 1—source data 1*) (*Rosen et al., 1981*; *Campbell et al., 1989*; *Russell et al., 1991*; *Yang et al., 2009*; *Lindsay et al., 1993b*; *Lindsay et al., 1993a*; *Lindsay et al., 1996*; *Lindsay et al., 2007*; *McManus et al., 1992*; *Merianos et al., 1992*; *Whelan et al., 1992*; *Whelan et al., 1995*, *Whelan et al., 1997*; *McDonnell et al., 1994*; *Russell, 1994*; *Russell, 2002*; *Dhileepan, 1996*; *Ritchie et al., 1997*; *Brokenshire et al., 2000*; *Ryan et al., 2000*; *Harley et al., 2000*; *Harley et al., 2001*; *Kelly-Hope et al., 2004a*; *Frances et al., 2004*; *Biggs and Mottram, 2008*; *Jacups et al., 2008b*; *Schmaedick et al., 2008*; *Lau et al., 2017*). *Ae. vigilax* and *Ae. notoscriptus* were more commonly implicated in transmission in tropical and subtropical zones, *Ae. camptorhynchus* in temperate zones, and *Cx. annulirostris* throughout all climatic zones. Freshwater-breeding *Cx. annulirostris* has been implicated in transmission across both inland and coastal areas, while saltmarsh mosquitoes *Ae. vigilax* and *Ae. camptorhynchus* have been implicated only in coastal areas (*Russell, 2002*) and inland areas affected by salinization from agriculture (*Biggs and Mottram, 2008*; *Carver et al., 2009*). Peri-domestic,

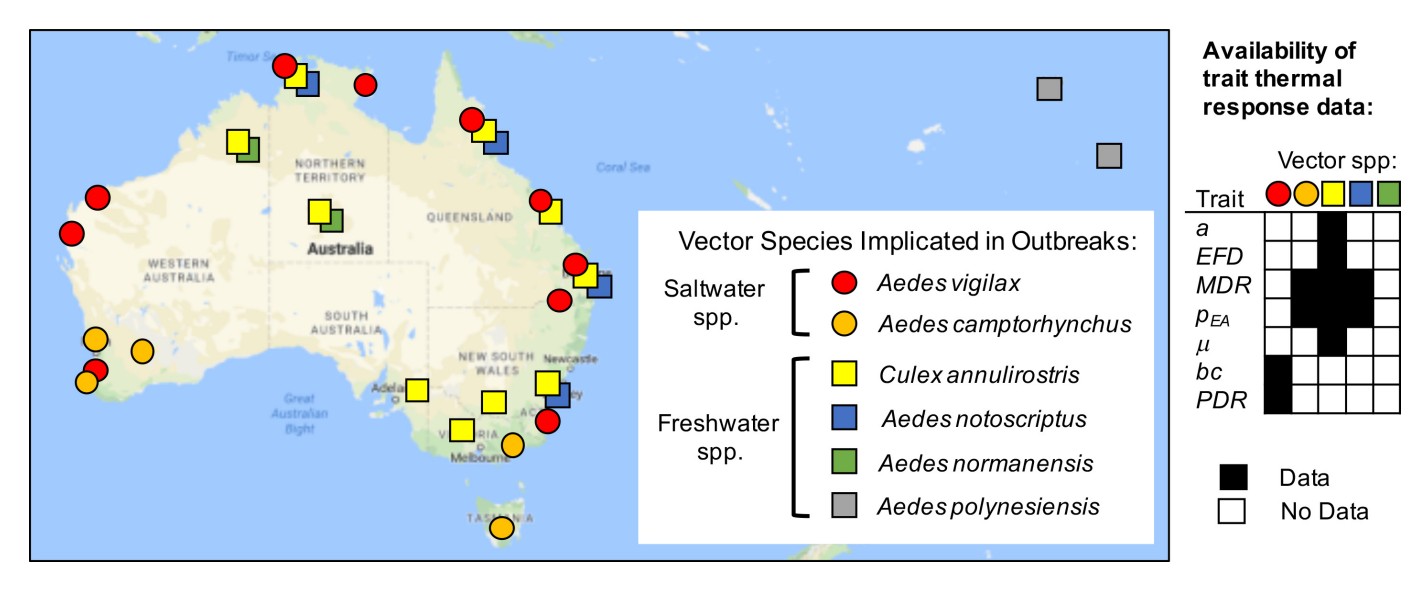

**Figure 1.** Vector species implicated in RRV disease outbreaks. Map of specific mosquito species identified as important vectors based on collected field specimens. Grid (right) shows data availability of trait thermal responses for the five Australian species. Data sources listed in *Figure 1—source data 1*. Trait parameters are biting rate (*a*), fecundity (as eggs per female per day, *EFD*), mosquito development rate (*MDR*), the proportion surviving from egg-to-adulthood ($p_{EA}$), adult mosquito mortality ($\mu = 1/lifespan$), vector competence (*bc*), and parasite development rate (*PDR*).
DOI: https://doi.org/10.7554/eLife.37762.003
The following source data is available for figure 1:

**Source data 1.** Vector species implicated in RRV disease outbreaks.
DOI: https://doi.org/10.7554/eLife.37762.004

container-breeding *Ae. notoscriptus* has been implicated in urban epidemics (*Russell, 2002*). The vectors also differ in their seasonality: *Ae. camptorhynchus* populations peak earlier and in cooler temperatures than *Ae. vigilax*, leading to seasonal succession where they overlap (*Yang et al., 2009*; *Russell, 1998*). This latitudinal and temporal variation suggests that vector species may have different thermal optima and/or niche breadths. If so, temperature may impact disease transmission differently for each species.

## General modeling approach

Transmission depends on a suite of vector, pathogen, and human traits, including mosquito density (*M*). Our main model ('full $R_0$ Model,' *Equation 1*) assumes temperature drives mosquito density and includes the relevant life history trait thermal responses (*Parham and Michael, 2009*; *Mordecai et al., 2013*; *Mordecai et al., 2017*). We initially compare this model to an alternative ('constant *M* model,' *Equation 2*) where mosquito density does not depend on temperature. We make this comparison because many transmission models do not include the thermal responses for mosquito density, assuming it depends primarily on habitat availability.

Here, we focus on the relative influence of temperature on transmission potential, recognizing that absolute $R_0$ also depends on other factors. Accordingly, we scaled model output between zero and one ('relative $R_0$'). Relative $R_0$ describes thermal suitability for transmission. Combined with factors like breeding habitat availability, vector control, humidity, human and reservoir host density, host immune status, and mosquito exposure, relative $R_0$ can be used to predict disease incidence. In this approach, only the relative thermal response of each trait influences $R_0$, which is desirable since traits can differ substantially due to other factors and in laboratory versus field settings (particularly mosquito survival: *Clements and Paterson, 1981*). Relative $R_0$ does not provide a threshold for sustained disease transmission (i.e. where absolute $R_0$ = 1), since this threshold is not controlled solely by temperature. Instead, relative $R_0$ preserves the temperature-dependence of $R_0$ to provide three key temperature values: upper and lower thermal limits where transmission is possible ($R_0 > 0$; a

conservative threshold where transmission is not excluded by high or low temperatures) and the temperature that maximizes $R_0$.

## Results

Vector and pathogen traits that drive transmission consistently responded to temperature (*Figure 2*), though data were sparse (*McDonald et al., 1980*; *Mottram et al., 1986*; *Russell, 1986*; *Rae, 1990*; *Kay and Jennings, 2002*). Although we exhaustively searched for experiments with trait measurements at three or more constant temperatures in the Australian vector species (*Cx. annulirostris*, *Ae. vigilax*, *Ae. camptorhynchus*, *Ae. notoscriptus*, and *Ae. normanensis*), no species had data for all necessary traits (*Figure 1*). Thus, we combined traits from two species to build composite $R_0$ models. We used mosquito life history traits measured in *Cx. annulirostris*: fecundity (as eggs per female per day, *EFD*), egg survival (as the proportion of rafts that hatch, *pRH*, and the number of larvae emerging per viable raft, *nLR*), the proportion surviving from larvae-to-adulthood ($p_{LA}$), mosquito development rate (*MDR*), adult mosquito lifespan (*lf*), and biting rate (*a*). We used infection traits measured in *Ae. vigilax*: vector competence (*bc*) and parasite development rate (*PDR*). For comparison, we also fit traits for other mosquito and virus species: *MDR* and $p_{LA}$ from *Ae. camptorhynchus* and *Ae. notoscriptus*, and *PDR* and *bc* from Murray Valley encephalitis virus (another important pathogen transmitted by these mosquitoes in Australia) in *Cx. annulirostris* (*Figure 2—figure supplements 2*, *3* and *4*) (*Kay et al., 1989*; *Barton and Aberton, 2005*; *Williams and Rau, 2011*). We used sensitivity analyses to evaluate the potential impact of this vector mismatch. However, all spatial and temporal predictions of $R_0$ (Figures 5–7) use the full $R_0$ model parameterized with mosquito life history traits from *Cx. annulirostris* and infection traits from *Ae. vigilax* (as shown in *Figure 2*).

Thermal optima ranged from 23.4°C for adult lifespan (*lf*) to 33.0°C for parasite development rate (*PDR*; *Figure 2*). The data supported unimodal thermal responses for most traits, though declines at high temperatures were not directly observed for biting rate (*a*) and parasite development rate. Data from other mosquito species and ectotherm physiology theory imply these traits must decline at very high temperatures, so we used strong priors to make them decline near ~40°C. Because our approach is designed to identify which traits constrain transmission at thermal limits, this choice is conservative since it means $R_0$ will be limited by other traits with better data. Accordingly, in the absence of data we preferred to overestimate upper thermal limits and underestimate lower thermal limits rather than vice versa.

Transmission potential (relative $R_0$ from the full $R_0$ model) peaked at 26.4°C, and was positive from 17.0–31.5°C (*Figure 3*). Removing the temperature-dependence of mosquito density [*M*] did not substantially affect the peak, because the optima for transmission and mosquito density were closely aligned (constant *M* model: 26.6°C, *M*: 26.2°C). By contrast, the range of temperatures suitable for transmission is much larger when mosquito density does not depend on temperature because *M*(*T*) constrains transmission at the thermal limits (constant *M* model positive from 12.9–33.7°C). The thermal constraints that mosquito density imposes on transmission are important because, although demographic traits are well-known to vary with temperature in the laboratory, many temperature-dependent transmission models do not assume that temperature influences mosquito density (*Martens et al., 1997*; *Craig et al., 1999*; *Paull et al., 2017*; *Caminade et al., 2017*; *Hamlet et al., 2018*, but see *Parham and Michael, 2009*; *Mordecai et al., 2013*, *Mordecai et al., 2017*; *Johnson et al., 2015*). The moderate optimal temperature for RRV (26–27°C) fits within the range of thermal optima found for other diseases: malaria transmission by *Anopheles* spp. at 25°C, and dengue and other viruses by *Ae. aegypti* and *Ae. albopictus* at 29 and 26°C, respectively (*Figure 4*) (*Mordecai et al., 2013*; *Mordecai et al., 2017*).

At the upper thermal limit fecundity (*EFD*) and adult lifespan (*lf*) constrain $R_0$, while at the lower thermal limit fecundity, larval survival (*pLA*), egg survival (raft viability [*pRH*] and survival within rafts [*nLR*]), and adult lifespan constrain $R_0$ (*Figure 3—figure supplement 2*). All of these traits (except adult lifespan) only occur in, and adult lifespan is quantitatively more important in, the full $R_0$ model, illustrating the importance of incorporating effects of temperature on vector life history. Correspondingly, uncertainty in these traits generated the most uncertainty in $R_0$ at the respective thermal limits (*Figure 3—figure supplement 2C*). The optimal temperature for $R_0$ was most sensitive to the thermal response of adult lifespan. Near the optimum, most uncertainty in $R_0$ was due to uncertainty in the thermal responses of biting rate and egg raft viability. For comparison, substituting larval traits

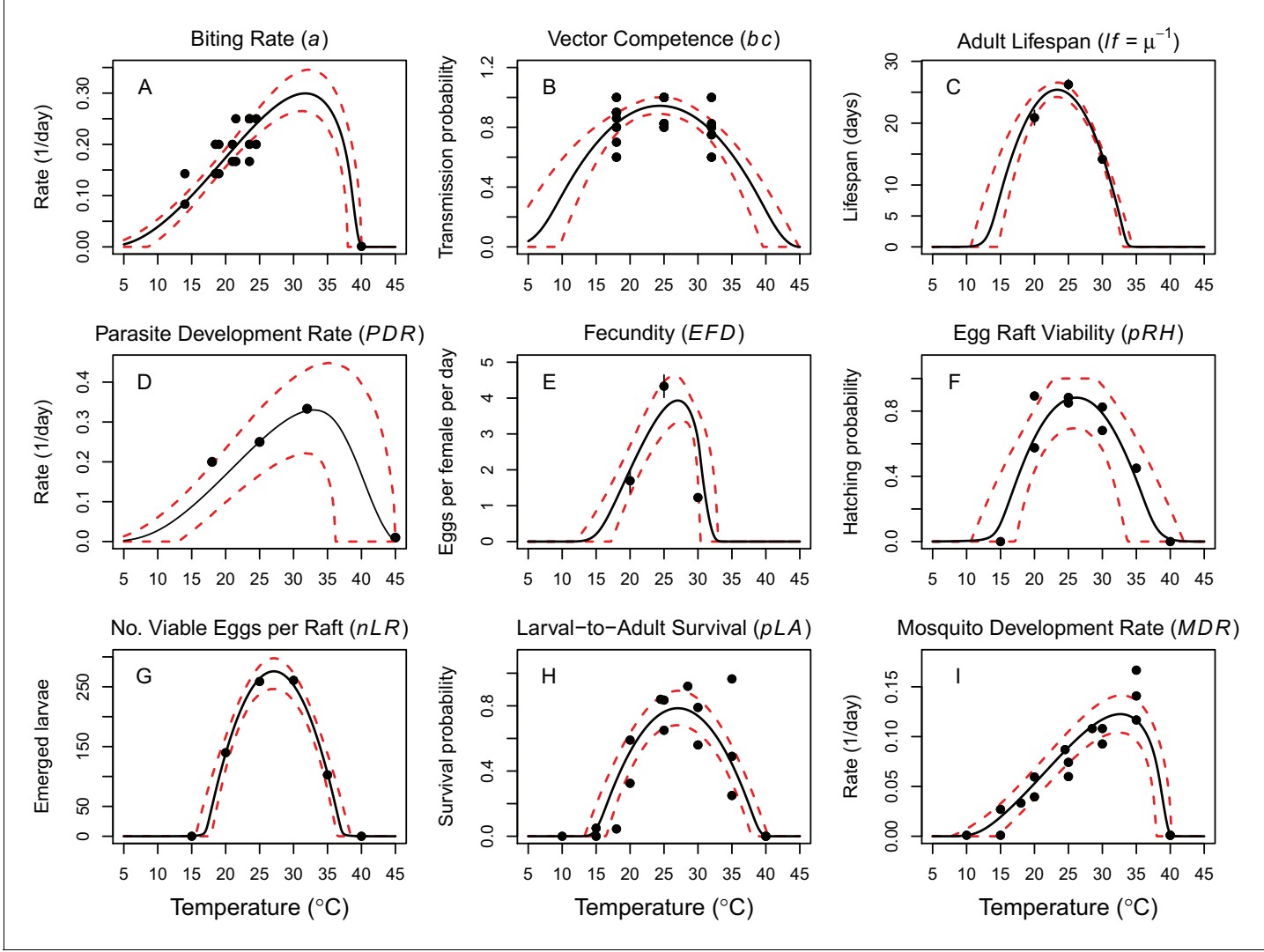

**Figure 2.** Thermal responses of *Cx. annulirostris* and RRV (in *Ae. vigilax*) traits that drive transmission. Mosquito life history traits (A, C, E, F, G, H, I) are from *Cx. annulirostris*. Virus-mosquito infection traits (B, D) are from *Ae. vigilax*. Functions were fit using Bayesian inference with priors fit using data from other mosquito species and viruses. Black solid lines are posterior distribution means; dashed red lines are 95% credible intervals. (E, C) Points are data means; error bars are standard error. Data sources and function parameter estimates given in *Figure 2—source data 1*. Data sources and function parameter estimates for priors given in *Figure 2—source data 2*. Thermal responses fit with uniform priors given in *Figure 2—figure supplement 1*. Thermal responses for alternative vectors and virus given in *Figure 2—figure supplements 2*, *3* and *4*.
DOI: https://doi.org/10.7554/eLife.37762.005

The following source data and figure supplements are available for figure 2:

**Source data 1.** Trait thermal response functions and data sources for Ross River virus $R_0$ models (*Equations 1 and 2*).
DOI: https://doi.org/10.7554/eLife.37762.011
**Source data 2.** Trait thermal response functions and data sources used to parameterize priors for data-informed trait thermal responses.
DOI: https://doi.org/10.7554/eLife.37762.012
**Figure supplement 1.** Thermal responses of *Cx. annulirostris* fit with uniform priors.
DOI: https://doi.org/10.7554/eLife.37762.006
**Figure supplement 2.** Thermal responses of *Ae. camptorhynchus*.
DOI: https://doi.org/10.7554/eLife.37762.007
**Figure supplement 2—source data 1.** Trait thermal response functions and data sources for Murray Valley Encephalitis virus and additional vector species (*Ae. notoscriptus* and *Ae. camptorhynchus*).
DOI: https://doi.org/10.7554/eLife.37762.008
**Figure supplement 3.** Thermal responses of *Ae. notoscriptus*.
DOI: https://doi.org/10.7554/eLife.37762.009
*Figure 2 continued on next page*

*Figure 2 continued*

**Figure supplement 4.** Thermal responses of Murray Valley Encephalitis virus.

DOI: https://doi.org/10.7554/eLife.37762.010

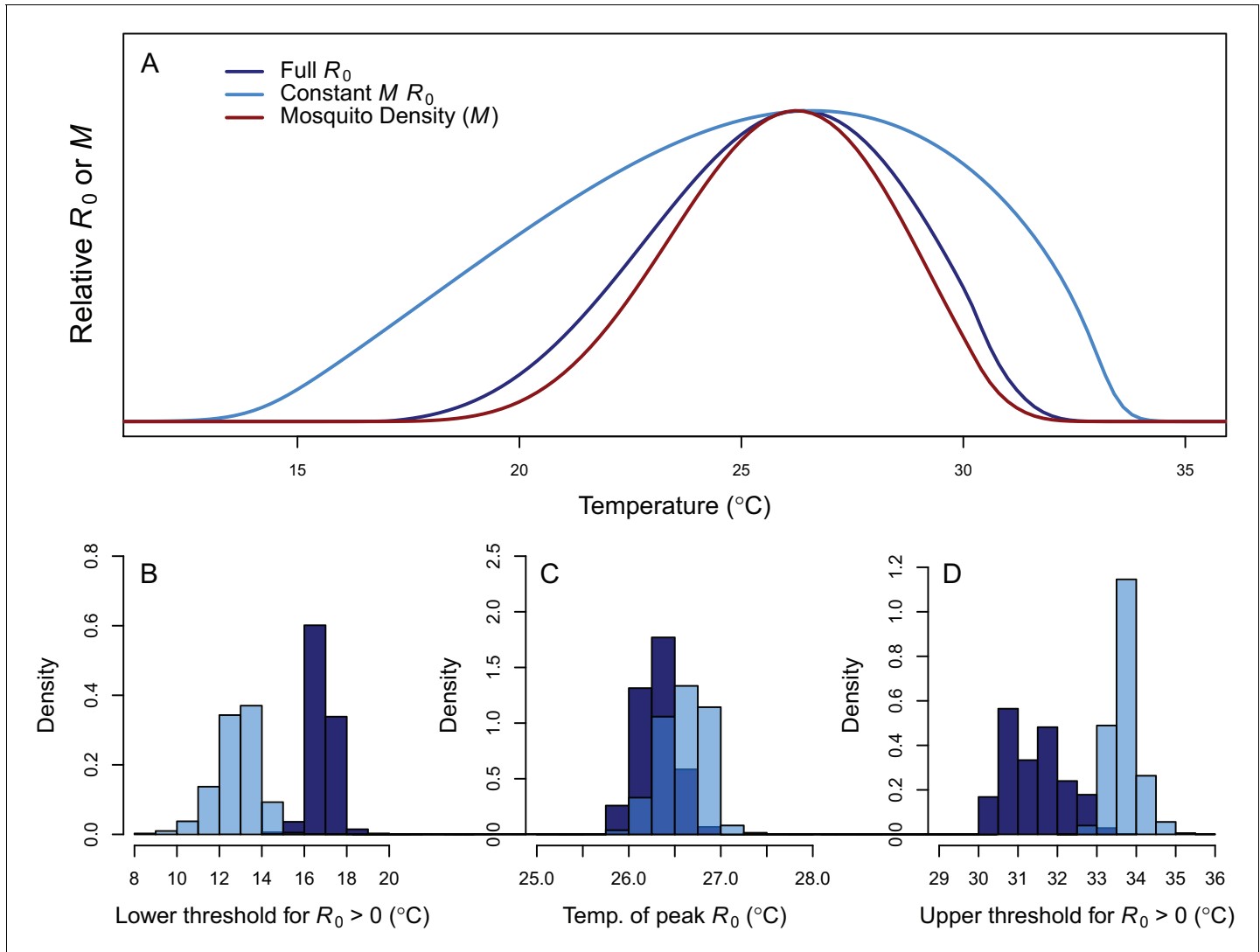

**Figure 3.** Thermal response of relative $R_0$. (A) Posterior means across temperature for the full $R_0$ model (*Equation 1*, dark blue) and constant $M$ model (*Equation 2*, light blue). Predicted mosquito density (M) shown for comparison (red). The y-axis shows relative $R_0$ (or $M$) rather than absolute values, which would require additional information. Histograms of posterior distributions for (B) critical thermal minimum, (C) thermal optimum, and (D) critical thermal maximum temperatures for both models (same colors as in A). Additional $R_0$ model results given in *Figure 3—figure supplement 1*. Sensitivity and uncertainty analyses given in *Figure 3—figure supplement 2*. Example comparison of mean and median results given in *Figure 3—figure supplement 3*.

DOI: https://doi.org/10.7554/eLife.37762.013

The following figure supplements are available for figure 3:

**Figure supplement 1.** Thermal response of relative $R0$ using traits from alternative vectors.

DOI: https://doi.org/10.7554/eLife.37762.014

**Figure supplement 2.** Sensitivity and uncertainty analyses for $R0$ results.

DOI: https://doi.org/10.7554/eLife.37762.015

**Figure supplement 3.** Example comparison of mean and median $R0$ results.

DOI: https://doi.org/10.7554/eLife.37762.016

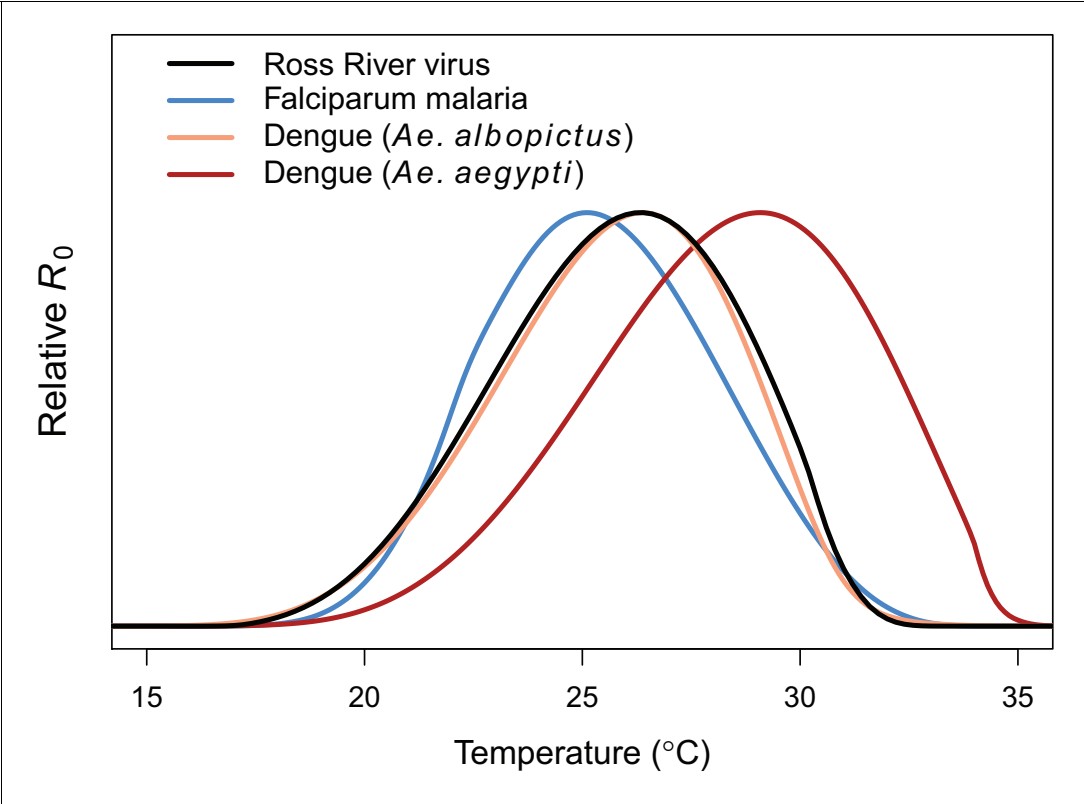

**Figure 4.** Comparing relative $R_0$ for RRV and other diseases. Malaria (blue, optimum = 25.2°C), Ross River virus (black, optimum = 26.4°C), dengue virus in *Ae. albopictus* (orange, optimum = 26.4°C), and dengue virus in *Ae. agypti* (red, optimum = 29.1°C). Results for all diseases use the full $R_0$ model.
DOI: https://doi.org/10.7554/eLife.37762.017

from alternative vectors or infection traits for Murray Valley Encephalitis virus did not substantially alter the $R_0$ thermal response, since *Cx. annulirostris* life history traits strongly constrained transmission (*Figure 3—figure supplement 1*).

Temperature suitability for RRV transmission varies seasonally across Australia, based on the full $R_0$ model (*Equation 1*) using monthly mean temperatures from WorldClim. In subtropical and temperate locations (Brisbane and further south), low temperatures force $R_0$ to zero for part of the year (*Figures 5A* and *6*). Monthly mean temperatures in these areas fall along the increasing portion of the $R_0$ curve for the entire year, so thermal suitability for transmission increases with temperature. By contrast, in tropical, northern Australia (Darwin and Cairns), the temperature remains suitable throughout the year (*Figures 5* and *6*). Darwin is the only major city where mean temperatures exceed the thermal optimum, and thereby depress transmission. Because most Australians live in southern, temperate areas, country-scale transmission is strongly seasonal. Using the average (1992–2013) seasonal incidence at the national scale, human cases peak two months after population-weighted $R_0(T)$, matching our *a priori* hypothesized time lag between temperature suitability and human cases (based on empirical work in other mosquito-borne disease systems, see Materials and methods and Discussion; *Figure 7*).

## Discussion

As the climate warms, it is critical to understand effects of temperature on transmission of mosquito-borne disease, particularly as new mosquito-borne pathogens emerge and spread worldwide. Identifying transmission optima and limits by characterizing nonlinear thermal responses, rather than simply assuming that transmission increases with temperature, can more accurately predict geographic, seasonal, and interannual variation in disease. Thermal responses vary substantially among diseases and vector species (*Mordecai et al., 2013*; *Mordecai et al., 2017*; Tesla et al., in press), yet we lack

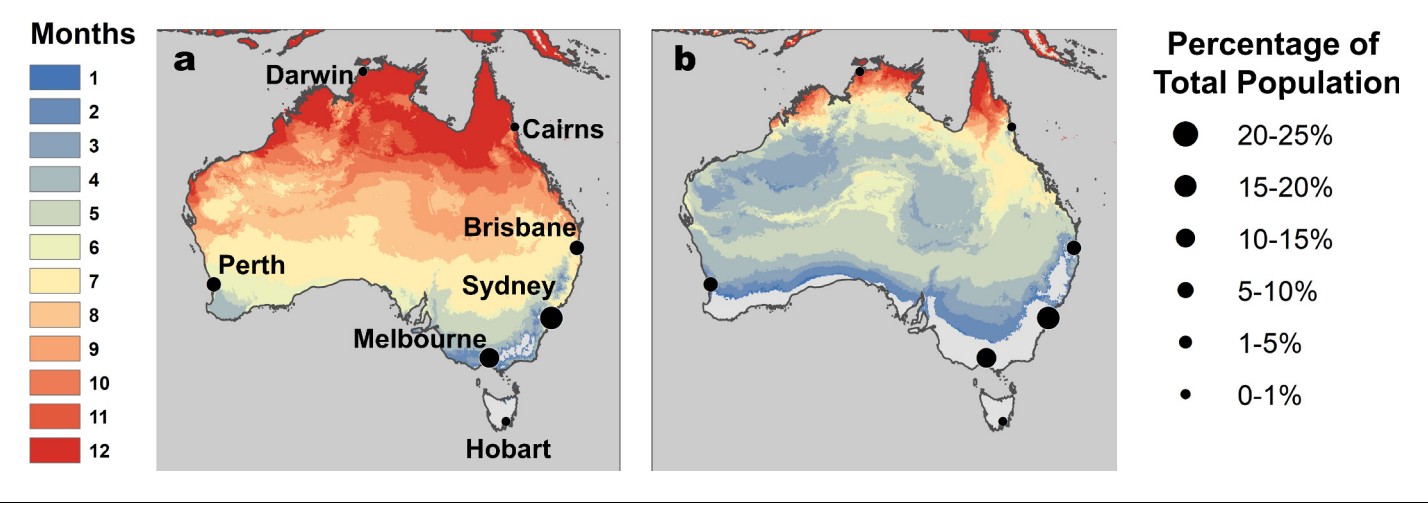

**Figure 5.** RRV transmission potential from monthly mean temperatures. Color indicates number of months where (**A**) relative $R_0$ >0 and (**B**) relative $R_0$ >0.5. Predictions are based on the posterior median of the full $R_0$ model (**Equation 1**) parameterized with trait thermal responses shown in **Figure 2**. Points indicate selected cities (**Figure 5**), scaled by the percentage of total Australian population residing in each city. Maps with 2.5% and 97.5% credible intervals are given in **Figure 5—figure supplement 1**.

DOI: https://doi.org/10.7554/eLife.37762.018

The following figure supplement is available for figure 5:

**Figure supplement 1.** RRV transmission potential from monthly mean temperatures using $R_0$ model 2.5% and 97.5% credible intervals.

DOI: https://doi.org/10.7554/eLife.37762.019

mechanistic models based on empirical, unimodal thermal responses for many diseases and vectors. Here, we parameterized a temperature-dependent model for transmission of RRV (**Figure 2**) with data from two important vector species (*Cx. annulirostris* and *Ae. vigilax*; **Figure 1**). The optimal temperature for transmission is moderate (26–27°C; **Figure 3**), and largely determined by the thermal response of adult mosquito lifespan (**Figure 3—figure supplement 2**). Both low and high temperatures limit transmission due to low mosquito fecundity and survival at all life stages—thermal responses that are often ignored in transmission models (**Figure 3—figure supplement 2**). Temperature explains the geography of year-round endemic versus seasonally epidemic disease (**Figures 5** and **6**) and accurately predicts the seasonality of human cases at the national scale (**Figure 7**). Thus, the model for RRV transmission provides a mechanistic link between geographic and seasonal variation in temperature and broad-scale patterns of disease.

While the thermal response of RRV transmission generally matched those of other mosquito-borne pathogens, there were some key differences. The moderate optimal temperature for RRV (26–27°C) fit within the range of thermal optima found for other diseases using the same methods: malaria transmission by *Anopheles* spp. at 25°C, and dengue and other viruses by *Ae. aegypti* and *Ae. albopictus* at 29 and 26°C, respectively (**Figure 4**) (**Mordecai et al., 2013**; **Mordecai et al., 2017**). For all of these diseases, the specific optimal temperature was largely determined by the thermal response of adult lifespan (**Mordecai et al., 2013**; **Mordecai et al., 2017**; **Johnson et al., 2015**). However, the traits that set the thermal limits for RRV transmission differed from other systems. The lower thermal limit for RRV was constrained by fecundity and survival at all stages, while the upper thermal limit was constrained by fecundity and adult lifespan. By contrast, thermal limits for malaria transmission were set by parasite development rate at cool temperatures and egg-to-adulthood survival at high temperatures (**Mordecai et al., 2013**). As with previous models, the upper and lower thermal limits of RRV transmission are more uncertain than the optimum (**Figure 3**) (**Johnson et al., 2015**; **Mordecai et al., 2017**), because trait responses are harder to measure near their thermal limits where survival is low and development is slow or incomplete. Overall, our results support a general pattern of intermediate thermal optima for transmission where the well-resolved optimal temperature is driven by adult mosquito lifespan, but upper and lower thermal limits are

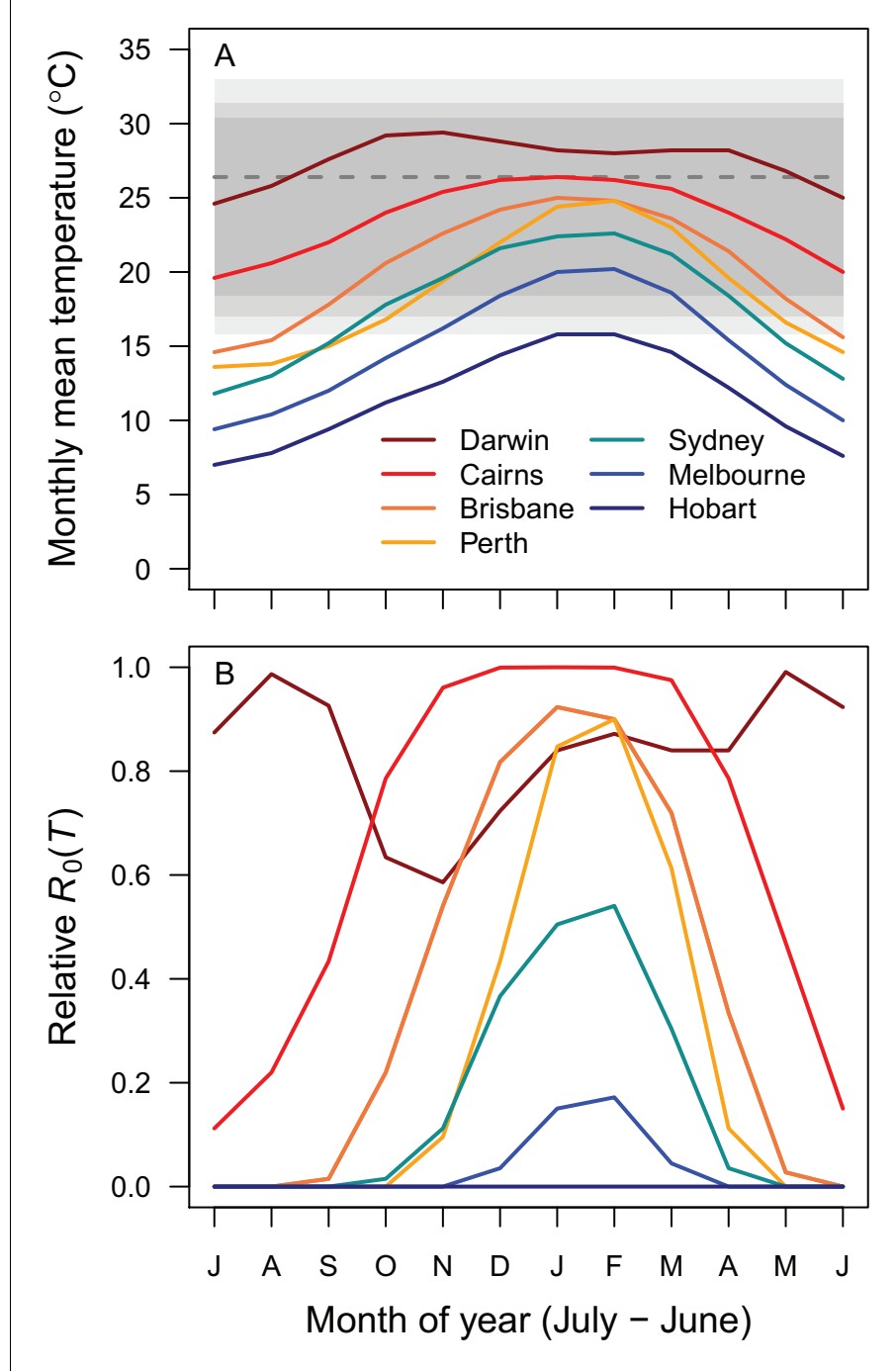

**Figure 6.** Average seasonality of temperature and relative $R_0$ in Australian cities. The selected cities span a latitudinal and temperature gradient (Darwin = dark red, Cairns = red, Brisbane = dark orange, Perth = light orange, Sydney = aqua, Melbourne = blue, Hobart = dark blue). The x-axis begins in July and ends in June (during winter). (**A**) Mean monthly temperatures. Shaded areas show temperature thresholds where $R_0 > 0$ for: outer 95% CI (light grey), median (medium grey), and inner 95% CI (dark grey). Dashed line shows median $R_0$ optimal temperature. (**B**) Temperature-dependent $R_0$. Predictions are based on the posterior median of the full $R_0$ model (*Equation 1*) parameterized with trait thermal responses shown in *Figure 2*.
DOI: https://doi.org/10.7554/eLife.37762.020

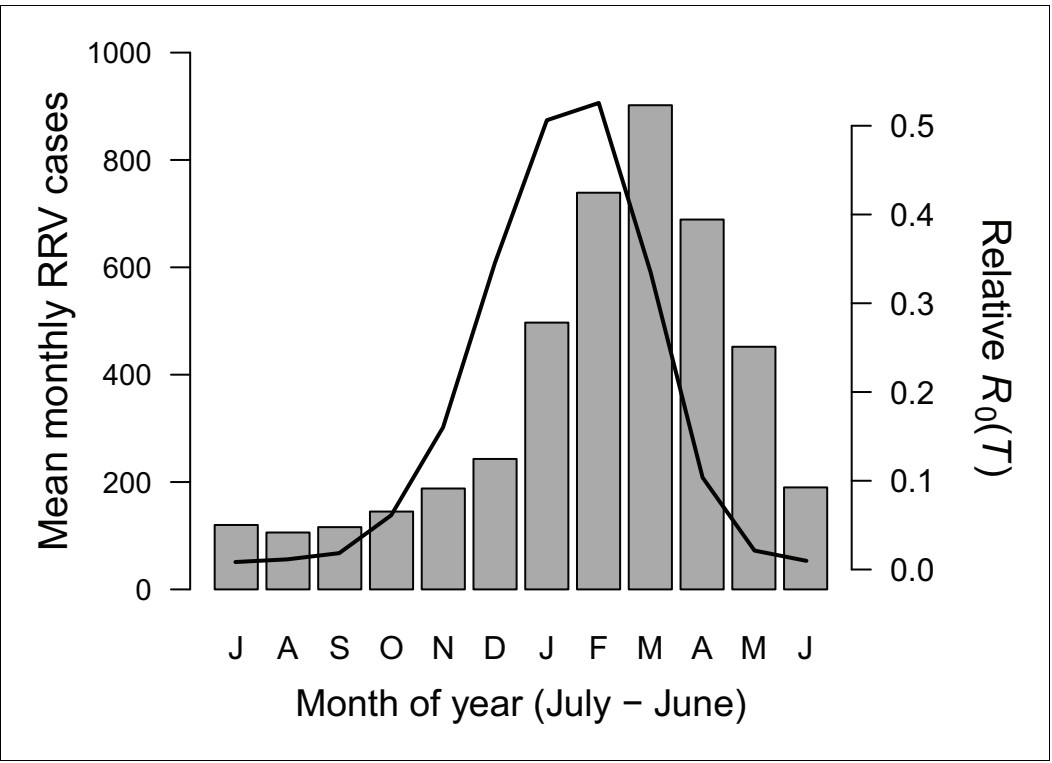

**Figure 7.** Seasonality of relative $R_0$ and RRV infections. Human cases aggregated nationwide from 1992 to 2013 (bars). Temperature-dependent $R_0$ weighted by population (line), calculated from Australia's 15 largest cities (76.6% of total population). Predictions are based on the posterior median of the full $R_0$ model (*Equation 1*) parameterized with trait thermal responses shown in *Figure 2*. The x-axis begins in July and ends in June (during winter). Cases peak two months after $R_0$, the *a priori* expected lag between temperature and reported cases.
DOI: https://doi.org/10.7554/eLife.37762.021

more uncertain and may be determined by unique traits for different vector–pathogen systems. Additionally, upper thermal limits of mosquito-borne disease transmission (a major concern for climate change) are primarily determined by vector life history traits with symmetrical thermal performance curves (like fecundity and survival at various life stages) rather than rate-based traits with asymmetrical thermal performance curves (like biting rate or pathogen development rate).

The trait thermal response data were limited in two keys ways. First, two traits (fecundity and adult lifespan) had data from only three temperatures. We used priors derived from data from other mosquito species to minimize over-fitting and better represent the true uncertainty (*Figure 2*, versus uniform priors in *Figure 2—figure supplement 1*). However, data from more temperatures would increase our confidence in the fitted thermal responses. Second, no vector species had data for all traits (*Figure 1*), so we combined mosquito traits from *Cx. annulirostris* and pathogen infection traits in *Ae. vigilax* to build composite relative $R_0$ models. Geographic and seasonal variation in vector populations suggests that *Ae. camptorhynchus* and *Ae. vigilax* have different thermal niches (cooler and warmer, respectively) and *Cx. annulirostris* has a broader thermal niche (*Figure 1*) (*Russell, 1998*). We need temperature-dependent trait data for more species to test the hypothesis that these niche differences reflect the species' thermal responses. If true, the current model, parameterized primarily with *Cx. annulirostris* trait responses, may not accurately predict the thermal responses of transmission by *Ae. camptorhynchus* and *Ae. vigilax*. Hypothesized species differences in thermal niche could explain why RRV persists over a wide climatic and latitudinal gradient. Thus, thermal response experiments with other RRV vectors are a critical area for future research.

The temperature-dependent $R_0$ model provides a mechanistic explanation for independently-observed patterns of RRV transmission across Australia. As predicted by the model (*Figures 5* and *6*), RRV is endemic in tropical Australia, with little seasonal variation in transmission potential due to

temperature, and seasonally epidemic in subtropical and temperate Australia (*Weinstein, 1997*). The model also accurately predicts disease seasonality at the national scale (*Figure 7*), reproducing the *a priori* predicted lag (8–10 weeks, or 2 months) for temperature to affect reported human cases (*Hu et al., 2006*; *Jacups et al., 2008b*; *Stewart Ibarra et al., 2013*; *Mordecai et al., 2017*). This lag between temperature and reported human cases arises from the time it takes for mosquito populations to increase, bite humans and reservoir hosts, acquire RRV, become infectious, and bite subsequent hosts; for pathogens to incubate with vectors; for humans to potentially develop symptoms, seek treatment, and report cases. Further, RRV transmission by *Cx. annulirostris* in inland areas often moves south as temperatures increase from spring into summer (*Russell, 1998*), matching the model prediction (*Figure 6*). Although temperature is often invoked as a potential driver for these types of patterns, it is difficult to establish causality from statistical inference alone, particularly if temperature and disease both exhibit strong seasonality and could both be responding to another latent driver. Thus, the mechanistic model is a critical piece of evidence linking temperature to patterns of disease.

In addition to explaining broad-scale patterns, the unimodal thermal model explains previously contradictory local-scale results. Specifically, statistical evidence for temperature impacts on local time series of cases is mixed. RRV incidence is often—but not always—positively associated with warmer temperatures (*Tong and Hu, 2001*; *Tong et al., 2002a*; *Tong et al., 2004*; *Hu et al., 2004*; *Hu et al., 2010*; *Jacups et al., 2008b*, *Williams et al., 2009*; *Werner et al., 2012*; *Koolhof et al., 2017*). However, variation in the effects of temperature on transmission across space and time is expected from an intermediate thermal optimum, especially when observed temperatures are near or varying around the optimum. The strongest statistical signal of temperature on disease is expected in temperate regions where mean temperature varies along the rapidly rising portion of the $R_0$ curve (~20–25°C). If mean temperatures vary both above and below the optimum (as in Darwin), important effects of temperature may be masked in time series models that fit linear responses. Additionally, if temperatures are always relatively suitable (as in tropical climates) or unsuitable (as in very cool temperate climates), variation in disease may be due primarily to other factors. A nonlinear mechanistic model is critical for estimating temperature impacts on transmission because the effect of increasing temperature by a few degrees can have a positive, negligible, or negative impact on $R_0$ along different parts of the thermal response curve. Although field-based evidence for unimodal thermal responses in vector-borne disease is rare (but see *Mordecai et al., 2013*; *Perkins et al., 2015*; *Peña-García et al., 2017*), there is some evidence for high temperatures constraining RRV transmission and vector populations: outbreaks were less likely with more days above 35°C in part of Queensland (*Gatton et al., 2005*) and populations of *Cx. annulirostris* peaked at 25°C and declined above 32°C in Victoria (*Dhileepan, 1996*). Future statistical analyses of RRV cases may benefit from using a nonlinear function for temperature-dependent $R_0$ as a predictor instead of raw temperature (*Figure 6B* versus *Figure 6A*).

Breeding habitat availability also drives mosquito abundance and mosquito-borne disease. Local rainfall or river flow have been linked to the abundance of RRV vector species (*Barton et al., 2004*; *Tall et al., 2014*; *Jacups et al., 2015*) and RRV disease cases (*Tong and Hu, 2001*; *Tong et al., 2002a*; *Hu et al., 2004*; *Kelly-Hope et al., 2004b*; *Tong et al., 2004*; *Gatton et al., 2005*; *Jacups et al., 2008b*; *Bi et al., 2009*; *Williams et al., 2009*; *Werner et al., 2012*), as have high tides in coastal areas with saltmarsh mosquitoes, *Ae. vigilax* and *Ae. camptorhynchus* (*Tong and Hu, 2002b*; *Tong et al., 2004*; *Jacups et al., 2008b*; *Kokkinn et al., 2009*). Overlaying models of species-specific breeding habitat with temperature-dependent models will better resolve the geographic and seasonal distribution of RRV transmission. Relative $R_0$ peaked at similar temperatures whether or not we assumed mosquito abundance was temperature-dependent (*Equation 1* versus *Equation 2*); however, the range of suitable temperatures was much wider for the model that assumed a temperature-independent mosquito population (*Figure 3*). Since breeding habitat can only impact vector populations when temperatures do not exclude them, it is critical to consider thermal constraints on mosquito abundance, even when breeding habitat is considered a stronger driver. Nonetheless, many mechanistic, temperature-dependent models of vector-borne disease transmission do not include thermal effects on vector density (*Martens et al., 1997*; *Craig et al., 1999*; *Paull et al., 2017*; *Caminade et al., 2017*; *Hamlet et al., 2018*). Our results demonstrate that the decision to exclude these relationships can have a critical impact on model results, especially near thermal limits.

Several important gaps remain in our understanding of RRV thermal ecology, in addition to the need for trait thermal response data for more vector species. First, the relative $R_0$ model needs to be more rigorously validated using time series of cases to determine the importance of temperature at finer spatiotemporal scales. These analyses should incorporate daily, seasonal, and spatial temperature variation, including aquatic larval habitat and adult microhabitat temperatures (*Paaijmans et al., 2010*; *Cator et al., 2013*; *Carrington et al., 2013*; *Thomas et al., 2018*). They should also integrate species-specific drivers of breeding habitat availability, like rainfall and tidal patterns, infrastructure (e.g. drainage), and human activities (e.g. deliberate and accidental water storage). Second, translating environmental suitability for transmission into human cases also depends on disease dynamics in reservoir host populations and their impact on immunity. For instance, in Western Australia heavy summer rains can fail to initiate epidemics when low rainfall in the preceding winter depresses recruitment of susceptible juvenile kangaroos (*Mackenzie et al., 2000*). By contrast, large outbreaks occur in southeastern Australia when high rainfall follows a dry year, presumably from higher transmission within relatively unexposed reservoir populations (*Woodruff et al., 2002*). Third, as the climate changes, long-term predictions should consider potential thermal adaption of vectors, since transmission at upper thermal limits is currently limited by vector life history traits. To date, we know very little about standing genetic variation for thermal performance or existing local thermal adaptation in vectors for any disease system. Building vector species-specific $R_0$ models and integrating thermal ecology with other drivers are important next steps for forecasting variation in RRV transmission. These more advanced models are necessary to translate our relative $R_0$ results into predictions of absolute $R_0$ (i.e. estimating the secondary cases per primary case, and where and when $R_0 > 1$ for sustained transmission).

Nonlinear thermal responses are particularly important for predicting how transmission will change under future climate regimes. Climate warming will likely increase the geographic and seasonal range of transmission potential in temperate, southern Australia where most Australians live. However, climate change will likely decrease transmission potential in tropical areas like Darwin, where moderate warming (~3°C) would push temperatures above the upper thermal limit for transmission for most of the year (*Figure 5*). However, the extent of climate-driven declines in transmission will depend on how much *Cx. annulirostris* and *Ae. vigilax* can adapt to extend their upper thermal limits and whether warmer-adapted vector species (e.g. *Ae. aegypti* and potentially *Ae. polynesiensis*) can invade and sustain RRV transmission cycles. Thus, we can predict the response of RRV transmission by current vector species to climate change based on these trait thermal responses. However, future disease dynamics will also depend on vector adaptation, potential vector species invasions, and climate change impacts on sea level and precipitation that drive vector habitat availability.

## Materials and methods

### Temperature-Dependent $R_0$ models

The 'full $R_0$ model' (*Equation 1*) assumes temperature drives mosquito density and includes vector life history trait thermal responses (*Parham and Michael, 2009*; *Mordecai et al., 2013*; *Mordecai et al., 2017*). The 'constant *M* model' (*Equation 2*) assumes mosquito density (*M*) does not depend on temperature (*Dietz, 1993*). There is disagreement in the literature over whether the equation for $R_0$ should contain the square root (*Dietz, 1993*; *Heffernan et al., 2005*; *Smith et al., 2012*). We use the version derived from the next-generation matrix method (*Dietz, 1993*) in order to be consistent with our previous work in other mosquito-borne disease systems (*Mordecai et al., 2013*; *Mordecai et al., 2017*; *Johnson et al., 2015*).

$$R_0(T) = \left( \frac{a(T)^2 bc(T) e^{-\frac{\mu(T)}{PDR(T)}} EFD(T) p_{EA}(T) MDR(T)}{N \, r \, \mu(T)^3} \right)^{1/2} \tag{1}$$

$$R_0(T) = \left( \frac{a(T)^2 bc(T) e^{-\frac{\mu(T)}{PDR(T)}} M}{N \, r \, \mu(T)} \right)^{1/2} \tag{2}$$

In both equations, (*T*) indicates that a parameter depends on temperature, *a* is mosquito biting

rate, *bc* is vector competence (proportion of mosquitoes becoming infectious post-exposure), μ is adult mosquito mortality rate (adult lifespan, *lf* = 1/μ), *PDR* is parasite development rate (*PDR* = 1/ *EIP*, the extrinsic incubation period), *N* is human density, and *r* is the recovery rate at which humans become immune (all rates are measured in days$^{-1}$). The latter two terms do not depend on temperature. In the full $R_0$ model, mosquito density (*M*) depends on fecundity (*EFD*, eggs per female per day), proportion surviving from egg-to-adulthood ($p_{EA}$), and mosquito development rate (*MDR*), divided by the square of adult mortality rate (μ) (*Parham and Michael, 2009*). We calculated $p_{EA}$ as the product of the proportion of egg rafts that hatch (*pRH*), the number of larvae per raft (*nLR*, scaled by the maximum at any temperature to calculate proportional egg survival within-rafts), and the proportion of larvae surviving to adulthood ($p_{LA}$).

We digitized previously published trait data (*Figure 2—figure supplement 1*; *McDonald et al., 1980*; *Mottram et al., 1986*; *Russell, 1986*; *Rae, 1990*; *Kay and Jennings, 2002*) using the free web-based tool Webplot Digitizer available at: https://automeris.io/WebPlotDigitizer/. We fit thermal responses of each trait using Bayesian inference with the 'r2jags' package (*Plummer, 2003*; *Su and Yajima, 2009*) in R (*R Core Team, 2017*). Traits with asymmetrical thermal responses were fit as Brière functions: $qT(T−T_{min})(T_{max}−T)^{1/2}$ (*Briere et al., 1999*). Traits with symmetrical thermal responses were fit as quadratic functions: $-q(T−T_{min})(T−T_{max})$. In both functions of temperature (*T*), $T_{min}$ and $T_{max}$ are the critical thermal minimum and maximum, respectively, and *q* is a rate parameter. For priors we used gamma distributions with hyperparameters derived from thermal responses fit to data from other mosquito species (*Figure 2—source data 2*) (*Davis, 1932*; *Jalil, 1972*; *McLean et al., 1974*; *Watts et al., 1987*; *Rueda et al., 1990*; *Focks et al., 1993*; *Joshi, 1996*; *Teng and Apperson, 2000*; *Tun-Lin et al., 2000*; *Alto and Juliano, 2001*; *Briegel and Timmermann, 2001*; *Kamimura et al., 2002*; *Calado and Navarro-Silva, 2002*; *Focks and Barrera, 2006*; *Wiwatanaratanabutr and Kittayapong, 2006*; *Lardeux et al., 2008*; *Delatte et al., 2009*; *Beserra et al., 2009*; *Yang et al., 2009*; *Westbrook et al., 2010*; *Muturi et al., 2011*; *Carrington et al., 2013*; *Tjaden et al., 2013*; *Eisen et al., 2014*; *Xiao et al., 2014*; *Ezeakacha, 2015*; *Morin et al., 2015*). These priors allowed us to more accurately represent the fit and uncertainty.

Our data did not include declining trait values at high temperatures for biting rate (*a*) and parasite development rate (*PDR*). Nonetheless, data from other mosquito species (*Mordecai et al., 2013*; *Mordecai et al., 2017*) and principles of thermal biology (*Dell et al., 2011*) imply these traits must decline at high temperatures. Thus, for those traits we included an artificial data point where the trait value approached zero at a very high temperature (40℃), allowing us to fit the Brière function. We used strongly informative priors to limit the effect of these traits on the upper thermal limit of $R_0$ (by constraining them to decline near 40℃). Because our approach is designed to identify which traits constrain transmission at thermal limits, this choice is conservative by allowing $R_0$ to be limited by other traits with better data. Accordingly, in the absence of data we favored overestimating $T_{max}$ and underestimating $T_{min}$ over the alternative. For comparison, we also fit all thermal responses with uniform priors (*Figure 2—figure supplement 1*); these results illustrate how the priors affected the results.

Bayesian inference produces estimated posterior distributions rather than a single estimated value. Because these distributions can be non-normal and asymmetric, we report and apply medians rather than means, since medians are less sensitive to outlying values in extended tails. However, we plot mean values in the figures because they show a smoother and more visually intuitive representation of where trait and $R_0$ thermal responses go to zero at the upper thermal limit. The means and medians are not substantially different, except at this thermal limit (see example in *Figure 3—figure supplement 3*).

## Sensitivity and uncertainty analyses

We conducted sensitivity and uncertainty analyses of the full $R_0$ model (*Equation 1*) to understand how trait thermal responses shape the thermal response of $R_0$. We examined the sensitivity of $R_0$ two ways. First, we evaluated the impact of each trait by setting it constant while allowing all other traits to vary with temperature. Second, we calculated the partial derivative of $R_0$ with respect to each trait across temperature ($\partial R_0/\partial X \cdot \partial X/\partial T$ for trait *X* and temperature *T*; *Equations 3-6*). To understand what data would most improve the model, we also calculated the proportion of total uncertainty in $R_0$ due to each trait across temperature. First, we propagated posterior samples from

all trait thermal response distributions through to $R_0(T)$ and calculated the width of the 95% highest posterior density interval (HPD interval; a type of credible interval) of this distribution at each temperature: the 'full $R_0(T)$ uncertainty'. Next, we sampled each trait from its posterior distribution while setting all other trait thermal responses to their posterior medians, and calculated the posterior distribution of $R_0(T)$ and the width of its 95% HPD interval across temperature: the 'single-trait $R_0(T)$ uncertainty'. Finally, we divided each single-trait $R_0(T)$ uncertainty by the full $R_0(T)$ uncertainty.

The partial derivatives are given below for all traits ($x$) that appear only once in the numerator of $R_0$ ($bc$, $EFD$, $pRH$, $nLR$, $pLA$, $MDR$; *Equation 3*), biting rate ($a$, *Equation 4*), parasite development rate ($PDR$, *Equation 5*), and lifespan ($lf$, *Equation 6*).

$$\frac{\partial R_0}{\partial x} = \frac{R_0}{2x} \tag{3}$$

$$\frac{\partial R_0}{\partial a} = \frac{R_0}{a} \tag{4}$$

$$\frac{\partial R_0}{\partial PDR} = \frac{R_0}{2\, lf\, PDR^2} \tag{5}$$

$$\frac{\partial R_0}{\partial lf} = \frac{R_0(1 + 3PDR)}{2\, PDR\, lf^2} \tag{6}$$

## Field observations: seasonality of temperature-dependent $R_0$ across Australia

We took monthly mean temperatures from WorldClim for seven cities spanning a latitudinal and temperature gradient (from tropical North to temperate South: Darwin, Cairns, Brisbane, Perth, Sydney, Melbourne, and Hobart) and calculated the posterior median $R_0(T)$ for each month at each location. We also compared the seasonality of a population-weighted $R_0(T)$ and nationally aggregated RRV cases. We used 2016 estimates for the fifteen most populous urban areas, which together contain 76.6% of Australia's population (*Australian Bureau of Statistics, 2017*). We calculated $R_0(T)$ for each location (as above) and estimated a population-weighted average. We compared this country-scale estimate of $R_0(T)$ with data on mean monthly human cases of RRV nationwide from 1992 to 2013 obtained from the National Notifiable Diseases Surveillance System.

We expected a time lag between temperature and reported human cases as mosquito populations increase, bite humans and reservoir hosts, acquire RRV, become infectious, and bite subsequent hosts; after an incubation period, hosts (potentially) become symptomatic, seek treatment, and report cases. Empirical work on dengue vectors in Ecuador identified a six-week time lag between temperature and mosquito oviposition (*Stewart Ibarra et al., 2013*). Subsequent mosquito development and incubation periods in mosquitoes and humans likely add another 2–4 week lag before cases appear, resulting in an 8–10 week lag between temperature and observed cases (*Hu et al., 2006*; *Jacups et al., 2008b*; *Mordecai et al., 2017*). With monthly case data, we hypothesize a two-month time lag between $R_0(T)$ and RRV disease cases.

## Mapping temperature-dependent $R_0$ across Australia

To illustrate temperature suitability for RRV transmission across Australia, we mapped the number of months for which relative $R_0(T) > 0$ and $> 0.5$ for the posterior median, 2.5 and 97.5% credibility bounds (*Figure 5—figure supplement 1*) for the full $R_0$ model (*Equation 1*). We calculated $R_0(T)$ at 0.2°C increments and projected it onto the landscape for monthly mean temperatures from WorldClim data at a 5 min resolution (approximately 10 km$^2$ at the equator). Climate data layers were extracted for the geographic area, defined using the Global Administrative Boundaries Databases (*GADM, 2012*). We performed map calculations and manipulations in R with packages 'raster' (*Hijmans, 2016*), 'maptools' (*Bivand and Lewin-Koh, 2017*), and 'Rgdal' (*Bivand et al., 2017*), and rendered GeoTiffs in ArcGIS version 10.3.1.

## Mosquito nomenclature

In 2000 there was a proposed shift in mosquito taxonomy: several subgenera within the genus *Aedes* were elevated to genus status (*Wilkerson et al., 2015*). This change affected *Aedes vigilax* and *Aedes camptorhynchus*, which were called *Ochlerotatus vigilax* and *Ochlerotatus camptorhynchus* for a time by some researchers. More recently, there has been a consensus to return to the previous naming system, so we use *Aedes* here, although many of the papers we cite use *Ochlerotatus* instead.

## Additional methods for digitizing trait data

The fecundity and adult survival data in *McDonald et al. (1980)* were published as time series of one experimental population at each temperature. The resulting data needed to be transformed to fit the corresponding trait thermal responses.

For survival, McDonald *et al.* reported the percent surviving approximately every other day (hereafter: 'semi-daily'). We used these data—along with the number of female adults alive on the first day of oviposition at each temperature—to generate a semi-daily time series estimating the number surviving. To generate the dataset that we used to directly fit the thermal responses, we converted this time series into the number of female individuals who died on each day (i.e. lifespan data).

For fecundity, McDonald *et al.* reported semi-daily fecundity data for entire population. Because the population was synchronized, and because mosquitoes lay discrete clutches of eggs separated by several days (the gonotrophic cycle duration), there were many data points when the populations did not produce offspring. These zero-inflated fecundity data are not ideal for fitting thermal responses. Therefore, after digitizing the semi-daily fecundity time series, we binned periods of several days (the bin size varied by temperature, since the gonotrophic cycle duration varies with temperature) and took a survival-weighted average within each bin (so days with more individual mothers contributing to offspring production counted more). To generate the dataset that we used to directly fit the thermal responses, we weighted the values within each bin by the mean number of surviving mothers in that bin. This approach allowed us to more accurately reflect daily fecundity averaged over a non-synchronized mosquito population. Note that the variation captured by these data and this approach is not variation between individual adult females, but rather variation by age for the whole population.

## Acknowledgements

We thank Leah Johnson and Matt Thomas for comments and Cameron Webb for posting the RRV case data (https://cameronwebb.wordpress.com/2014/04/09/why-is-mosquito-borne-disease-risk-greater-in-autumn/). All authors were supported by the National Science Foundation (DEB-1518681; https://nsf.gov/). EAM was supported by the NSF (DEB-1640780; https://nsf.gov/), the Stanford Woods Institute for the Environment (https://woods.stanford.edu/research/environmental-venture-projects), and the Stanford Center for Innovation in Global Health (http://globalhealth.stanford.edu/research/seed-grants.html).

## Additional information

### Funding

| Funder | Grant reference number | Author |
| --- | --- | --- |
| National Science Foundation | DEB-1518681 | Erin A Mordecai |
| National Science Foundation | DEB-1640780 | Erin A Mordecai |

The funders had no role in study design, data collection and interpretation, or the decision to submit the work for publication.

### Author contributions

Marta Strecker Shocket, Data curation, Formal analysis, Investigation, Visualization, Methodology, Writing—original draft, Writing—review and editing; Sadie J Ryan, Data curation, Formal analysis,

Investigation, Visualization, Writing—review and editing; Erin A Mordecai, Conceptualization, Resources, Supervision, Funding acquisition, Methodology, Project administration, Writing—review and editing

### Author ORCIDs
Marta Strecker Shocket  http://orcid.org/0000-0002-8995-4446
Sadie J Ryan  http://orcid.org/0000-0002-4308-6321

### Decision letter and Author response
Decision letter https://doi.org/10.7554/eLife.37762.026
Author response https://doi.org/10.7554/eLife.37762.027

---

# Additional files

### Supplementary files
• Transparent reporting form
DOI: https://doi.org/10.7554/eLife.37762.022

### Data availability
All data and R code used for analyses in this study are included in the supporting files and also available via Dryad (http://dx.doi.org/10.5061/dryad.m0603gk).

The following dataset was generated:

| Author(s) | Year | Dataset title | Dataset URL | Database, license, and accessibility information |
|---|---|---|---|---|
| Shocket M, Ryan S, Mordecai E | 2018 | Data from: Temperature explains broad patterns of Ross River virus transmission | http://dx.doi.org/10.5061/dryad.m0603gk | Available at Dryad Digital Repository under a CC0 Public Domain Dedication |

---

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
