## [Decision Letter]

Thank you for submitting your article "Temperature explains broad patterns of Ross River virus transmission" for consideration by *eLife*. Your article has been reviewed by two peer reviewers, and the evaluation has been overseen by a Reviewing Editor and Ian Baldwin as the Senior Editor. The reviewers have opted to remain anonymous.

The reviewers have discussed the reviews with one another and the Reviewing Editor has drafted this decision to help you prepare a revised submission.

Summary:

Another very nice manuscript by Mordecai and colleagues. Their approach has worked successfully for better understanding malaria, dengue and Zika and chikungunya transmission, and is now applied to another yet underappreciated infectious disease: Ross River virus (with a focus on Australia).

Essential revisions:

1) The authors highlight on various occasions that there is quite a bit of uncertainty in the data, given that good reliable experimental data are scarce. One way of solving the uncertainly at the high temperature end is to use the so-called ectotherm physiology theory, which implies traits must decline at very high temperature (40°C).

However, looking at Figure 2 the data for the various traits do not seem to match (biologically). For example, adults do not survive at temperatures of 34°C and beyond. Yet, at these very same temperatures they continue to bite. Should those traits not be capped at the same temperature as well (34°C)? Same at lower temperature extreme. Or perhaps it is the other way round: lifespan is 0 days at 40°C?

2) Mean monthly temperatures are used to create the suitability maps, but how well do these temperatures represent the various vector breeding habitats and available adult mosquito resting sites?

3) Just wondering: looking at your *PDR* curve and your mosquito curves, do you hypothesize that the virus is already prepared for climate change, and that the limiting factor or bottleneck is the mosquito at moment? If true, subtle differences in the upper limit (and/or the ability to adapt) between mosquito species may become important.

4) The comparison of the results to data on seasonality is welcome but lacks two key components. First, the authors do not describe how the timing of the RRV peak corresponds to that of the model-based estimates. They state 'as hypothesized', but do not relate this hypothesis which is critical to the results. They also characterize this finding as 'matching […] the seasonal peak' in the Abstract, which is not explicitly the case because there is a delay. Second, The authors only make a single comparison despite having a data on multiple outbreaks in multiple parts of the country and highlighting their region specific estimates. The density model for Australia is spatiotemporal should it should be analyzed against spatiotemporal data (e.g. Discussion, fourth paragraph), not just summarized temporal data.

5) The way the mosquito species are combined in the final model is unclear, e.g. Figure 5. Some data were available from different species and were used in different combinations for different components, but I could not tell which ones entered the final model or why that is expected to be a reasonable model despite the differences between species which the authors highlight at multiple points (e.g. Discussion, last paragraph). Uncertainty around this is compounded by the limited validation (i.e. comment above).

6) The Discussion should include limitations of the utility of the results due to estimating 'relative *R_0_*' as opposed to *R_0_*.

7) The authors included a square root in the *R_0_* calculation which changes its traditional meaning. It's not clear why they have made this modification and it probably has little if any impact on the results. Nonetheless, it would be best to use the traditional equation to reduce confusion in the literature.

---

## [Author Response]

Essential revisions:1) The authors highlight on various occasions that there is quite a bit of uncertainty in the data, given that good reliable experimental data are scarce. One way of solving the uncertainly at the high temperature end is to use the so-called ectotherm physiology theory, which implies traits must decline at very high temperature (40°C).However, looking at Figure 2 the data for the various traits do not seem to match (biologically). For example, adults do not survive at temperatures of 34°C and beyond. Yet, at these very same temperatures they continue to bite. Should those traits not be capped at the same tempertaure as well (34°C)? Same at lower temperature extreme. Or perhaps it is the other way round: lifespan is 0 days at 40°C?

The reviewers are correct identifying this apparent contradiction. Because it is difficult to measure traits at biologically extreme temperatures where survival and organismal performance are low, the shape of thermal performance curves at lower and upper thermal limits is always somewhat uncertain. To ensure that our models were not artificially constrained by assumptions about trait performance at temperatures for which they were not measured, we conservatively forced these traits to zero only at the extreme temperature of 40°C. In the *R_0_* model, transmission is constrained by the traits with the most restrictive *T_min_* and *T_max_*. Therefore, the positive biting rate at temperatures where adult survival is zero does not result in any predicted transmission. Given the lack of biting rate data at higher temperatures, we did not want biting rate to artificially constrain transmission (i.e., in the absence of data it is much better to overestimate *T_max_* and underestimate *T_min_* than vice versa). Additionally, we consulted with our colleague Dr. Matthew Thomas (an empirical mosquito physiologist at Penn State) whose expert opinion (based on years of measuring thermal performance curves in mosquitoes) is that biting rate is the last trait to go to zero – if mosquitoes are alive, they will bite. He argues that biting rate will never constrain *R_0_*, and that models of transmission should reflect that.

We revised the manuscript (in the Materials and methods and in the Results) to make the rationale behind our approach clearer.

2) Mean monthly temperatures are used to create the suitability maps, but how well do these temperatures represent the various vector breeding habitats and available adult mosquito resting sites?

The reviewers are correct that water stores heat differently than air, and therefore larval habitat temperatures do not exactly match the surrounding air temperatures. (The discrepancy between air and water temperatures depends on the volume of the body of water.) Thermal consequences of adult mosquito microhabitat selection is an open area of research. However, our suitability maps are already so coarse in scale (monthly mean temperatures) that these differences are likely trivial compared to other sources of temperature variation that we also do not include (i.e., daily and weekly thermal variation). Nonetheless, our coarse-scale, constant-temperature analysis is an important and necessary first step before it is possible to incorporate finer-scale and varying temperature analyses.

We added text (in the Discussion) to point out these additional sources of temperature variation as limitations on our results and important avenues for further study.

3) Just wondering: looking at your PDR curve and your mosquito curves, do you hypothesize that the virus is already prepared for climate change, and that the limiting factor or bottleneck is the mosquito at moment? If true, subtle differences in the upper limit (and/or the ability to adapt) between mosquito species may become important.

The reviewers are correct. For almost all organisms, development rates peak at higher temperatures than other life history traits like survival and fecundity. Thus, these traits – and particularly adult lifespan – are the key limitation on disease transmission at high temperatures. Moreover, differences in the thermal response of lifespan between mosquito species are very important (e.g., Mordecai et al., 2017 predict a 3°C difference in the optimal temperature for dengue transmission by *Ae. aegypti* and *Ae. albopictus* that arises from differences in mosquito survival at high temperatures). As a result, any evolutionary response of mosquito lifespan to climate change will impact how climate change affects disease transmission.

We revised the manuscript (in the Discussion) to better emphasize these points.

4) The comparison of the results to data on seasonality is welcome but lacks two key components. First, the authors do not describe how the timing of the RRV peak corresponds to that of the model-based estimates. They state 'as hypothesized', but do not relate this hypothesis which is critical to the results. They also characterize this finding as 'matching… the seasonal peak' in the Abstract, which is not explicitly the case because there is a delay. Second, The authors only make a single comparison despite having a data on multiple outbreaks in multiple parts of the country and highlighting their region specific estimates. The density model for Australia is spatiotemporal should it should be analyzed against spatiotemporal data (e.g. Discussion, fourth paragraph), not just summarized temporal data.

The reviewers make an excellent point. Our model estimates the value of *R_0_* at constant temperatures but does not simulate transmission dynamically in a population. Therefore, our model cannot directly predict when human cases should peak as temperature changes over time, since temporal lags are introduced based on the time it takes for mosquito abundance to increase, the virus to incubate in the vectors, transmission to new humans to occur, and for humans to present symptoms, seek healthcare, and have cases recorded. Our “predicted seasonality” is based on independent work in other systems showing that there is often a 6-week lag between temperature and mosquito population abundance, resulting in a 8-10 week lag between temperature and observed cases. Based on this predicted lag (2 months), the seasonality of human cases is consistent with our temperature-based *R_0_* model.

We agree that a spatiotemporal analysis is an important and obvious next step. However, we do not currently have multi-year or multi-location outbreak data. If we are able to obtain these data, we hope to accomplish this spatiotemporal analysis in a future paper. Although we do not have sufficient data to fully validate the temperature-dependent *R_0_* model, the seasonal RRV data shows that our model is consistent with broad-scale observed patterns of disease. Validating this type of model (where the output is relative *R_0_* that depends only on temperature) with human incidence data across time and space is non-trivial and there are no consensus methods for doing so. We are currently working on developing new methods to validate the model (with the spatiotemporal data described above that we hope to obtain). However, doing so is beyond the scope of this article.

We have revised our description of the seasonality results to make the rationale for the expected two-month lag between peak *R_0_* and peak cases clearer (in the Introduction, Results, and Discussion).

5) The way the mosquito species are combined in the final model is unclear, e.g. Figure 5. Some data were available from different species and were used in different combinations for different components, but I could not tell which ones entered the final model or why that is expected to be a reasonable model despite the differences between species which the authors highlight at multiple points (e.g. Discussion, last paragraph). Uncertainty around this is compounded by the limited validation (i.e. comment above).

We have added text to the manuscript (in the Results, and in various figure captions [Figure 2; Figure 5; Figure 6; Figure 7]) to be clearer about which species traits are incorporated into the main *R_0_* model. As in Figure 2, mosquito-only traits are from *Cx. annulirostris* and infection traits are from *Ae. vigilax*. The trait fits from other vector and virus species (and the corresponding alternative *R_0_* models) are only shown in supplemental figures for comparison with the main model.

The main text *R_0_* model is primarily constrained by *Cx. annulirostris* traits, and is a reasonable model for transmission by *Cx. annulirostris* because the temperature-dependence of vector transmission is largely driven by vector traits (Mordecai et al., 2013, 2017). For other mosquito species, including *Ae. vigilax*, demographic and infection rate traits are likely to vary, resulting in different relationships between *R_0_* and temperature. A major aim of our research program and this paper is to rigorously and clearly identify what trait data from which vector species are available in the literature, and to highlight how that impacts our predictions for how disease transmission will respond to climate change. There is a common misconception that the relationships between vector and pathogen traits and temperature are already well-known, yet that is clearly not the case. For these reasons, we hope this work will motivate further empirical work on the thermal responses of other Australian vector species, an important research gap that we highlight in the Discussion.

6) The Discussion should include limitations of the utility of the results due to estimating 'relative R_0_' as opposed to R_0_.

The reviewer raises a very important discussion point. Our major goals in this paper are to synthesize the multiple vector and parasite traits by which temperature influences transmission, to predict the overall response of transmission to temperature, and to identify the traits that drive both the response of transmission to temperature and the uncertainty in this response. To accomplish these goals, we needed a metric for transmission that incorporates the (nonlinear) influence of multiple traits, which can be parameterized with laboratory experiment data in which thermal responses can be measured rigorously. Relative *R_0_* is an appropriate metric for this purpose because it incorporates the known temperature-dependent processes (and their nonlinear responses) involved in vector transmission. In this and previous papers, we have shown that relative *R_0_* models parameterized in this way can accurately predict the temperature response of transmission in the field (Mordecai et al., 2013, 2017). However, our relative *R_0_* metric differs from some traditional uses *R_0_* as the number of secondary cases arising from a primary case in a fully susceptible population (i.e., the absolute reproductive number), and to calculate the critical vaccination threshold. This departure is intentional, because we do not expect temperature to be the only driver of variation in *R_0_*, nor do we expect laboratory experiments to capture *quantitative* transmission rates in the field. Instead, we are most interested in the *relative* effect of temperature on transmission potential, the optimal temperature for transmission, and the thermal limits at which transmission potential declines to zero.

We now more fully describe this rationale (third and last paragraphs in the Introduction) and explicitly mention translating relative *R_0_* into absolute *R_0_* (seventh paragraph in the Discussion).

7) The authors included a square root in the R_0_ calculation which changes its traditional meaning. It's not clear why they have made this modification and it probably has little if any impact on the results. Nonetheless, it would be best to use the traditional equation to reduce confusion in the literature.

We recognize that there are multiple forms of *R_0_* commonly used in the literature, and that their meaning and derivation differ (Dietz 1993, Heffernan et al., 2005, Smith et al., 2012). As we explain in response 6, we are only using *R_0_* to predict relative effects of temperature on transmission and to identify the thermal limits and optima. These metrics are not affected by whether our equation or its square is used because there is a monotonic, 1:1 relationship between the two forms (we previously demonstrated this in the supplementary materials of Mordecai et al., 2017, Figure F). The method we use here was derived from the next generation matrix method and is used in our previous analyses for malaria (Mordecai et al., 2013), dengue (Mordecai et al., 2017), and Zika (Tesla et al. bioRxiv, 2018). By keeping the model methods consistent, we facilitate comparison across systems.

We have added text indicating that there is disagreement in the literature and providing our reasoning (subsection “Temperature-Dependent R0 Models”, first paragraph in the Materials and methods).